# Singlet Oxygen in Photodynamic Therapy

**DOI:** 10.3390/ph17101274

**Published:** 2024-09-26

**Authors:** Shengdong Cui, Xingran Guo, Sen Wang, Zhe Wei, Deliang Huang, Xianzeng Zhang, Timothy C. Zhu, Zheng Huang

**Affiliations:** 1MOE Key Laboratory of Medical Optoelectronics Science and Technology, Key Laboratory of Photonics Technology of Fujian Province, School of Optoelectronics and Information Engineering, Fujian Normal University, Fuzhou 350117, Chinahyjsuki@yeah.net (S.W.);; 2Department of Radiation Oncology, University of Pennsylvania, Philadelphia, PA 19104, USA

**Keywords:** singlet oxygen, photodynamic therapy, photosensitizer, detection methods

## Abstract

Photodynamic therapy (PDT) is a therapeutic modality that depends on the interaction of light, photosensitizers, and oxygen. The photon absorption and energy transfer process can lead to the Type II photochemical reaction of the photosensitizer and the production of singlet oxygen (^1^O_2_), which strongly oxidizes and reacts with biomolecules, ultimately causing oxidative damage to the target cells. Therefore, ^1^O_2_ is regarded as the key photocytotoxic species accountable for the initial photodynamic reactions for Type II photosensitizers. This article will provide a comprehensive review of ^1^O_2_ properties, ^1^O_2_ production, and ^1^O_2_ detection in the PDT process. The available ^1^O_2_ data of regulatory-approved photosensitizing drugs will also be discussed.

## 1. Introduction

Photodynamic therapy (PDT) is a modern therapeutic technology that depends on the interdisciplinary integration of pharmaceutics, optics, and medicine. PDT has been used to treat various benign and malignant diseases [1]. The therapeutic responses of PDT depend on the interaction of light, photosensitizers (PSs), and molecular oxygen within the target cells and tissues [2]. In general, PDT involves two main steps: the topical or systemic administration of photosensitizing drugs and light irradiation of the treatment site accumulated with the PS. In the presence of molecular oxygen, this process leads to the generation of reactive oxygen species (ROS) and, particularly, the photosensitized formation of singlet oxygen (^1^O_2_), the key photophysical step in Type II photosensitization. The initial biological responses and effectiveness of PDT treatment are ultimately determined by the temporal and spatial distribution of ^1^O_2_.

Photosensitizers commonly employed in clinical practice can be classified into groups such as porphyrins, chlorins, or other dyes from the chemist’s point of view. Porphyrins are a widely used PS, which include Porfimer Sodium, PhotoGem, and Hiporfin [3], also known as hematoporphyrin derivative (HpD), which is the complex mixture of water-soluble porphyrin monomers and oligomers that are purified from animal blood [4]. Aminolevulinic acid (ALA) and its ester derivatives are prodrugs that are metabolized in cells and enzymatically converted to protoporphyrin IX (PpIX), a potent endogenous porphyrin type PS [5]. Chlorin-type PSs used in clinical practice includes Temoporfin (m-THPC) and Laserphyrin (LS11) [6,7]. Phthalocyanines and organic dyes, for instance, aluminum phthalocyanine tetrasulfonate (ALPcS4), usually have long absorption wavelengths and high extinction coefficients [8].

Most PSs exhibit PDT effects primarily characterized by the Type II photochemical reaction and the efficient production of ^1^O_2_ [9]. ^1^O_2_ strongly oxidizes and reacts with biomolecules. The mechanism of ^1^O_2_-mediated PDT highly depends on the subcellular localization of the PS and the oxidative destruction of organelles, such as mitochondria, lysosomes, endoplasmic reticulum, Golgi apparatus, and plasma membranes. When exceeding the cell’s antioxidation capacity, the acute oxidative stress to these organelles can induce changes in calcium and lipid metabolism, the production of cytokines and stress proteins, and the activation of protein kinases and transcription factors. These cellular responses can ultimately lead to apoptosis and necrosis of targeted eukaryotic cells and prokaryotic cells. Subsequently, ^1^O_2_-generated cytotoxicity exerts a distinctive therapeutic mechanism of the direct killing of target cells, damaging blood vessels, and inducing an immune effect [10]. Therefore, ^1^O_2_ is regarded as the key photocytotoxic species accountable for the initial tissue damage during PDT. The triplet sensitization in the PDT process is a very effective way to generate ^1^O_2_, which makes ^1^O_2_-mediated PDT a powerful treatment modality with unique mechanisms and broad clinical applications [11]. This article provides a comprehensive review focusing on the ^1^O_2_ production from a clinically used PS based on the recent literature.

## 2. Properties of Singlet Oxygen

### 2.1. Electronic Structure and Leaps in Singlet Oxygen

According to the molecular orbital theory, the two highest energy electrons of triplet-state oxygen should be filled in two π-antibonding orbitals and spin-parallel, respectively. It has two unpaired electrons; thus it is paramagnetic. This oxygen molecule with unpaired electrons is triplet-state oxygen, symbolized as ^3^O_2_ (^3^$\sum_{g}^{-}$) [12]. Figure 1 illustrates the three lowest electronic state energy level transitions and molecular orbital schematics of oxygen molecules. When the ^3^O_2_ becomes excited, the two electrons with opposite spins and the highest energy can be arranged in two ways: (1) The two electrons with opposite spins occupy a single π orbital simultaneously, and the other orbital is the empty orbital, which is the first excited singlet state of the oxygen molecule, i.e., singlet oxygen, symbolized as ^1^O_2_ (^1^Δ_g_). (2) The two electrons with opposite spins each occupy two π orbitals, which is the second excited singlet state oxygen molecule, symbolized as ^1^O_2_ (^1^$\sum_{g}^{+}$) [13]. The three forms of the oxygen molecule are called spin isomers: ^3^O_2_ (^3^$\sum_{g}^{-}$), ^1^O_2_ (^1^Δ_g_), and ^1^O_2_ (^1^$\sum_{g}^{+}$). The energy difference between ^1^O_2_ (^1^$\sum_{g}^{+}$) and ^1^O_2_ (^1^Δ_g_) is 0.56 eV, while the energy difference between ^1^O_2_ (^1^Δ_g_) and ^3^O_2_ (^3^$\sum_{g}^{-}$) is 0.98 eV [14]. The decay of ^1^O_2_ (^1^$\sum_{g}^{+}$) to ^1^O_2_ (^1^Δ_g_) by the internal conversion process is fast, while ^1^O_2_ (^1^Δ_g_) to ^3^O_2_ (^3^$\sum_{g}^{-}$) is a slower process. So, in general, singlet oxygen refers to the ^1^O_2_ (^1^Δ_g_) state [13].

Salokhiddinov et al. report that the radioluminescence intensity of transition (5) (λ = 1270 nm) is 60 times higher than that of transition (4) (λ = 1590 nm), based on the detection of the ^1^O_2_ luminescence signal of mesoporphyrin IX dimethyl ester dissolved in carbon tetrachloride (CCl_4_) using a highly sensitive germanium diode detector [15]. Macpherson et al. report that the radioluminescence intensity of transition (5) is 100 times higher than that of transition (4), based on the study of the ^1^O_2_ luminescence signals of 5,10,15,20-tetra(4-sulfonato) phenylporphine (TTPS), 5,10,15,20-tetrakis (pentafluorophenyl) porphine (TPPF), and 5,10,15,20-tetraphenylporphine (TTP) in different solutions (e.g., methanol, toluene, CCl_4_, etc.) [16].

Since the energy (ΔE) of ^1^O_2_ (^1^Δ_g_) is 0.98 eV higher than that of ^3^O_2_ (^3^$\sum_{g}^{-}$), the difference in excitation energy from the S_0_ to the S_1_ and the difference in energy transfer from the T_1_ to the S_0_ of the PS should be higher than 0.98 eV, which is a necessary but not sufficient condition for the production of ^1^O_2_ by the PS. Spiller et al. suggest that the excitation energy difference of S_0_ to S_1_ (see Figure 2) is 1.85 eV for phthalocyanine zinc(II) (ZnPc) and 1.63 eV for 1,4,8,11,15,18,22,25-octa (hexyloxy) phthalo cyanine zinc(II) (ZnOHP), respectively [17]. Darwent et al. suggest that the T_1_ to S_0_ energy transfer differences for 2,9,16,23-tetrasulfophthalocyanine zinc(II) (ZnPTC) and 5,10,15,20-tetraphenyl porphyrin zinc(II) (ZnTPP) are 1.12 eV and 1.59 eV, respectively [18].

### 2.2. Physical Chemistry Properties of Singlet Oxygen

Without reacting with the surrounding molecules, ^1^O_2_ can return to the ground state and emit photons through spontaneous radiation transition, known as ^1^O_2_ luminescence (or phosphorescence), corresponding the wavelength (*λ*) of an emission peak near 1270 nm [19]. But not all ^1^O_2_ produced emits light through spontaneous radiation. The luminescence rate constant (*k_e_*) for ^1^O_2_ is the rate at which ^1^O_2_ is consumed by the luminescence process upon return to the ground state. The *k_e_* of ^1^O_2_ varies in different solvents, e.g., the *k_e_* of ^1^O_2_ in water is 0.25 s^−1^ [20]. The intrinsic lifetime of ^1^O_2_ refers to its lifetime in an ideal state without undergoing any chemical reactions, and it is described by τ_∆_ = 1/*k_e_*. τ_∆_ is dependent on the properties of molecular structure and energy level. In experimental conditions, the lifetimes of ^1^O_2_ can vary in different solutions, cellular environments, or biological environments. Table 1 lists the lifetimes of ^1^O_2_ in different solvents.

The lifetime of ^1^O_2_ is mainly determined by the quenching rate constant (*k_q_*), which falls under two categories: physical and chemical. Physical quenching involves energy transfer without a chemical reaction. In this process, ^1^O_2_ interacts with a quenching agent and transfers its energy to heat or other forms of energy before returning to ^3^O_2_. For instance, excited singlet-state *β*-carotene can transfer energy to ^1^O_2_, producing triplet-state *β*-carotene and ^3^O_2_, therefore, effectively quenching 250~1000 molecules of ^1^O_2_ [21]. The chemical quenching process involves a reaction in which a chemical reacts with the quencher to create a new compound, and ultimately leads to a change in the chemical structure of the quencher or even results in quencher depletion. For example, ^1^O_2_ reacts with amines to produce free radicals and ascorbate to form hydrogen peroxide (H_2_O_2_) [21,22]. The quenching effect reduces the concentration of ^1^O_2_. The rate of quenching depends on the stereoelectronic structure of the quencher molecule. For example, C-H, N-H, and O-H groups can quickly transfer vibrational energy, a non-radiative transition process, and can distribute the excited molecular energy through the vibrational modes within the molecule, thus speeding up the quenching process [21]. The quenching rate directly impacts the lifetime of ^1^O_2_. A fast quenching process shortens the lifetime, whereas a slow process or low quencher concentration prolongs it. The *k_q_*, *k_e_*, and τ_∆_ of porphyrins are listed in Table 1.

**Table 1 pharmaceuticals-17-01274-t001:** The quenching rate constant, luminescence rate constant, and lifetime of ^1^O_2_ in different solvents.

Solvent	*k_q_ *(M^−1^s^−1^)	*k_e_* (M^−1^s^−1^)	τ_∆_ (μs)
H_2_O	2.9 × 10^9 a^	0.25 ^b^	3.1 ^a,b,c^
CH_3_OH	1.2 × 10^9 c^	0.81 ^b^	9.5 ^a^
C_6_H_14_	5.5 × 10^8 a^	N/A	23.4 ^a^
C_6_H_6_	3.9 × 10^8 a^	4.5 ^b^	30 ^a^
C_5_H_12_	4.5 × 10^8 a^	N/A	34.7 ^a^
D_2_O	3.7 × 10^8 a^	0.22 ^b^	68 ^a,b^
CD_3_OD	N/A	0.79 ^b^	270 ^b^

^a^: data from [23]; ^b^: data from [20]; and ^c^: data from [24]. N/A: not available.

Quenching can be achieved by adding a quenching agent, which is now mostly used to verify the existence of ^1^O_2_. The use of sodium azide (NaN_3_) in an aqueous solution at a concentration greater than 9 mM can quench ^1^O_2_, and adding deuterium oxide (D_2_O) possibly also extends the lifetime of ^1^O_2_ to make detection easier. Table 2 summarizes the *k_q_* and the quenching mode of several commonly used ^1^O_2_ quenchers.

Cells contain various biomolecules, including lipids, proteins, and nucleic acids. These biomolecules can react rapidly with ^1^O_2_ and cause quenching. Lipids play essential roles as a fundamental part of the cell membrane, a source of energy storage, and as an intermediary in various signaling processes. The reaction of ^1^O_2_ with the free and esterified forms of unsaturated fatty acids in lipid molecules mainly involves para-reactions, resulting in isomeric allylic hydroperoxides. ^1^O_2_ also reacts with cholesterol to produce cholesterol aldehydes [30]. ^1^O_2_ reacts with several amino acid residues in proteins (e.g., histidine, tryptophan, and tyrosine) in a reaction that is essentially a chemical quenching of ^1^O_2_. Therefore, ^1^O_2_ has a short lifetime and limited diffusion distance within the cell. Moan et al. demonstrate that the average diffusion distance of ^1^O_2_ in the cellular environment is 100~150 nm, and it is 20 nm intracellularly with the ^1^O_2_ lifetime of 40 ns in the skin and liver of living rats [31]. Ethirajan et al. suggest that the diffusion distance of ^1^O_2_ in the cellular environment is only 45 nm, as the lifetimes in the lipid and cytoplasmic regions of the cell membrane are 100 ns and 250 ns, respectively [32]. These data suggest that ^1^O_2_ cannot penetrate through the cellular membrane.

### 2.3. Oxidizing Activity of Singlet Oxygen

The oxidizing capacity can be measured by its standard redox potential. The standard redox potential of ^1^O_2_ is 2.23 V, which is 1 V higher than that of ^3^O_2_ [33]. H_2_O_2_, a Type I ROS, is also a strong oxidizer with a standard redox potential of 1.77 V [34]. Although H_2_O_2_ is a two-electron strong oxidizer, it exhibits minimal or no interaction with the majority of biomolecules, including low molecular weight antioxidants (e.g., vitamin E and vitamin C) [35]. The presence of a pair of electrons with opposite spins in the highest occupied molecular orbital confers dienophile features to ^1^O_2_. ^1^O_2_ readily reacts with unsaturated organic compounds by electrophilic addition and electron extraction [13]. It can also react with free radicals through energy transfer or electron transfer. In certain chemical reactions, ^1^O_2_ can transfer energy to other molecules, resulting in the creation of free radicals. For instance, ^1^O_2_ reacts with unsaturated fatty acids in cell membranes, leading to the formation of hydroperoxides through Alder-ene reactions with linoleic and linolenic acids, and ultimately causing lipid peroxidation [36]. The oxidation of amino acid residues in proteins by ^1^O_2_ (e.g., the oxidation of methionine residues to sulfoxides, oxidation of histidine to hydroxyimidazolone, and oxidation of tryptophan to N-formylkynurenine) leads to protein structural changes and function loss [37,38]. ^1^O_2_ can also react with lipids or thiols via electron transfer, leading to free radical formation [39,40]. ^1^O_2_ also induces the oxidation of guanine and causes oxidative damage of DNA [41]. These reactions and the high redox potential suggest that the oxidizing power of ^1^O_2_ is stronger than H_2_O_2_. However, the decomposition of H_2_O_2_ creates super-oxidizing hydroxyl radicals (^•^OH) with an oxidation potential of 2.8 V, making them potent oxidizing ROS produced in the PDT process [42]. This can also be reflected by the difference between the wide ranges of the half-life of different ROS. For instance, the half-life of ^1^O_2_ ranges between 10^−9^ and 10^−6^ s, and that of ^•^OH between 10^−9^ and 10^−7^ s [43].

Antioxidants are substances that, when present in low concentrations compared to those of an oxidizable substrate, significantly inhibit or prevent the oxidation of that substrate. They act as reducing agents, donating electrons to neutralize free radicals, thereby preventing the formation of oxidative chain reactions that can damage cells and lead to various diseases and conditions [44]. An imbalance between oxidants and antioxidants in a biological system favors oxidants, which can lead to damage known as oxidative stress [45]. Both light-dependent and light-independent processes in biological systems may produce ^1^O_2_. The light-independent reactions include those catalyzed by peroxidases or oxygenases, and reactions of H_2_O_2_ with hypochlorite (ClO^−^) or nitrite. Additionally, the recombination of peroxyl radicals (ROO^•^) derived from biomolecules can lead to the release of ^1^O_2_ [30]. Additionally, light-dependent processes undergo photosensitization to generate ^1^O_2_. Experimental evidence has directly or indirectly suggested that the following naturally occurring ROS are major ones causing oxidative damage in the human body: superoxide radical anion ($O_{2}^{•-}$), H_2_O_2_, ROO^•^, ^•^OH, ^1^O_2_, and peroxynitrite (ONOO^−^) [46]. To counteract the assault of these ROS, living cells have a biological defense system composed of enzymatic antioxidants that convert ROS to harmless species. For example, $O_{2}^{•-}$ can be converted to ^3^O_2_ and H_2_O_2_ by superoxide dismutase, and H_2_O_2_ can be converted to water and ^3^O_2_ [46]. In contrast, some ROS are dependent on quenching by various non-enzymatic antioxidants. Non-enzymatic endogenous antioxidants include lipid-based antioxidants (e.g., carotenoids and vitamin E) and water-soluble antioxidants (e.g., vitamin C, glutathione, and hemoglobin) [47].

Excess ^1^O_2_ in biological systems can destroy cells or tissues, potentially causing irreversible oxidative damage, carcinogenesis, and promoting tumor survival. When the antioxidant system is dysregulated, such as through the overexpression of glutamate–cysteine ligase and glutamate–cysteine catalytic subunit, it can lead to an imbalance in the glutathione/glutathione disulfide ratio [48], generating an excess of ^1^O_2_. This imbalance can disrupt the integrity of biomolecules, cause cellular damage, generate oxidative stress, and damage mitochondria and DNA, ultimately leading to apoptosis, necrosis, senescence, and a proliferation blockade [49].

Physiological levels of ^1^O_2_ have signaling roles within the cell and may act as modulators of cell signaling [49]. It exerts a significant influence on cellular activities, usually with reversibly oxidized hydrogen sulfide groups as signaling molecules to maintain a dynamic equilibrium of generated and quenched ^1^O_2_. The amount of ^1^O_2_ in an organism is regulated by various factors, including the physiological state and environmental conditions. Nevertheless, it is difficult to define a high or low level of ^1^O_2_ as a specific value.

## 3. Singlet Oxygen Production in PDT Process

PDT is a two-step procedure: (1) the patient is first topically or systemically given a PS (“photosensitizing drug”) that preferentially accumulates in the target tissue, and (2) the light of a specific wavelength generated by a laser light source, LED light source, or daylight is then applied to the treatment site in order to active the PS. The process of PDT to produce ^1^O_2_ involves the interaction of PS molecules with light irradiation in the presence of molecular oxygen. Therefore, the PS, light, and oxygen are considered as the three crucial elements of PDT. The process of ^1^O_2_ production includes photon absorption, emission, and energy transfer as the PS undergoes different electronic states to eventually generate an excited triplet-state PS and convert the absorbed light energy to highly cytotoxic ^1^O_2_. PSs used in PDT are usually conjugated unsaturated organic molecules that possess high light absorption coefficients, high intersystem crossing efficiencies, and high quantum yields for the production of ^1^O_2_ or other ROS [50]. 

When the PS is exposed to the light of a specific wavelength and absorbs photon energy, PS molecules in the ground state (S_0_) are transitioned to an energy level above the first excited singlet state (S_1_). Subsequently, the PS molecules in the S_1_ will undergo vibrational relaxation and either decay to the lowest vibrational energy level of that electronic state or decay through internal conversion and vibrational relaxation to the lowest energy level of S_1_. 

The lifetime of S_1_ is short (10^−8^~10^−9^ s) due to the rapid rate of internal transitions [51]. Therefore, the observed luminescence of an S_1_ radiation, accompanied by the S_1_ to S_0_ radiative transitions, is called fluorescence. PS molecules in the S_1_ undergo intersystem crossing (ISC) through non-radiative transitions from the S_1_ to the slightly lower energy excited triplet state (T_1_). The emission of light resulting from the transition from the T_1_ to S_0_ is known as phosphorescence. This progress is spin-forbidden, leading to a low rate constant, with a lifetime ranging from 10^−4^ to 10^2^ s for the T_1_ [51].

Because the S_1_ of the PS molecule has a much shorter lifetime than the T_1_, the likelihood of energy transfer with ^3^O_2_ is reduced. The T_1_-state PS, with its longer lifetime, can effectively transfer energy to ^3^O_2_, leading to the formation of ^1^O_2_. A simplified Jablonski energy level diagram for the Type I and Type II photochemical reactions in the PDT process is shown in Figure 2.

The concentration of ^1^O_2_ generated during PDT is a crucial factor in determining the therapeutic effect, and its concentration is related to many factors and can be expressed as the following:(1)O21t=NδS0ΦΔτΔτT−τΔexp−tτT−exp−tτΔ
where [^1^O_2_] represents the concentration of ^1^O_2_ at time t, N is the number of photons during the excitation pulse, δ the absorption cross-section of the PS, [S_0_] is the concentration of the ground-state PS, Φ_∆_ is the quantum yield of ^1^O_2_, τ_T_ is the lifetime of the PS in its triplet state, and τ_Δ_ is the lifetime of ^1^O_2_.

## 4. Singlet Oxygen Detection

### 4.1. Direct Detection of Singlet Oxygen

Detection methods for ^1^O_2_ are broadly categorized into direct and indirect methods. The direct detection of ^1^O_2_ luminescence is considered the “gold standard” for PDT dosimetry. ^1^O_2_ is directly detected by measuring its luminescence at 1270 nm [52]. The long wavelength and low energy of ^1^O_2_, along with its low emission probability in biological media (approximately 10^−8^), contribute to its inactivation through collision with water molecules under physiological conditions. This results in a short lifetime of ^1^O_2_ on solvents or biological environments, rendering its accurate detection exceedingly arduous using direct detection methods. Therefore, near-infrared photodetectors based on high sensitivity, when integrated with time-resolved counting techniques (e.g., gated photon counting, multichannel counting, and time-correlated single-photon counting), have emerged as a pivotal method for the direct detection of ^1^O_2_ lifetimes. Currently, high-sensitivity NIR detectors for the direct detection of ^1^O_2_ luminescence include the photomultiplier tube (PMT) [53], InGaAs/InP single photon avalanche diode (SPAD) [54], and superconducting nanowire single photon detector (SNSPD) [55,56].

### 4.2. Indirect Detection of Singlet Oxygen

#### 4.2.1. Electron Paramagnetic Resonance

The electron paramagnetic resonance (EPR) method detects ^1^O_2_ by utilizing the change in the EPR signal after the EPR spin-trapping agent interacts with the ^1^O_2_. The method can determine the yield of ^1^O_2_ by detecting the change in the EPR signal. EPR has a high degree of sensitivity and selectivity in the detection of ^1^O_2_, but it is susceptible to the interference of solvents and coexisting ions. In 1976, McIntosh and Bolton combined the phenomenon of chemically induced dynamic electron polarization with changes in the rate of free radical generation to specifically detect the production of ^1^O_2_ as low as 100 nM [57]. Many researchers have tried to improve the EPR method for the study of ^1^O_2_. For instance, to study the ^1^O_2_ quenching effect of hydroxyl radicals in the presence of thioredoxin by using 2,2,6,6,-tetramethyl-4-piperidine (TMP) as a trapping agent [58]. However, EPR spectroscopy is not ideal for detecting intracellular ^1^O_2_, given that its temporal resolution typically spans from microseconds to milliseconds, whereas the lifetime of ^1^O_2_ within cells is usually in the nanosecond scale [12].

#### 4.2.2. Fluorescence Photometry

Fluorescent probes are characterized by high sensitivity, simple data acquisition, high imaging resolution, etc., and are excellent sensors for detecting ^1^O_2_ [59]. By detecting changes in fluorescence properties (fluorescence at specific wavelengths, changes in fluorescence signal intensity, fluorescence quantum yield, etc.), it is possible to determine whether or not ^1^O_2_ is produced. Commonly used organic fluorescent probes mainly include probes with fluorescent signals of their own, which have a significant change in fluorescent signal intensity after ^1^O_2_ is captured, and probes that do not have fluorescent signals of their own, but will generate strong fluorescent signal after ^1^O_2_ is captured. The fluorescence probe includes 1,3 Diphenylisobenzofuran (DPBF) [60], 9-[2-(3-Carboxy-9,10-diphenyl)anyhryl]-6-hydroxy-3H-xanthen-3-ones (DPAXs) [61], and 9-[2-(3-carboxy-9,10-dimethyl)anyhryl]-6-hydroxy-3H-xanthen-3-one (DMAX) [62]. DMAX reacts faster with ^1^O_2_ and is 53 times more sensitive than DPAXs. While DPBF (9.6 × 10^8^ M^−1^s^−1^) exhibits a higher rate constant for its reaction with ^1^O_2_ in water environments compared to DMAX (9.1 × 10^8^ M^−1^s^−1^), its reactivity with ClO^−^ and hydroxyl radicals diminishes its selectivity as a probe for ^1^O_2_ detection [63]. These probes react with ^1^O_2_ to form stable endoperoxides. They have been used to detect ^1^O_2_ in neutral or alkaline aqueous solutions. 

The singlet oxygen green sensor (SOSG) is a fluorescent probe for in vitro detection that is highly selective for ^1^O_2_ [64]. To apply the fluorescent probe to biological samples, the fluorescent probe needs to penetrate the cell membrane and localize inside the cell to directly react with the ^1^O_2_ molecules. Recently, 9,10-dimethylanthracene (DMA) and silicon-containing rhodamine (Si-rhodamine) moieties, a far-red fluorescent probe namely Si-DMA, have been developed to detect ^1^O_2_ in the mitochondria. Murotomi et al. demonstrated that Si-DMA can quantitatively measure intracellular ^1^O_2_ in living cells [65].

#### 4.2.3. Spectrophotometry

Spectrophotometry utilizes an absorption probe to detect ^1^O_2_, where the absorbance value changes upon capture of the ^1^O_2_ and the difference in absorbance is used to express the amount of ^1^O_2_ produced. Spectrophotometry is a relatively simple method for the detection of ^1^O_2_, and the commonly used absorption probes include 9,10-anthracenedipro-pionicacid (ADPA) [66] and 9,10-diphenyl anthracene (DPA) [67]. Some probes can dissolve in water and undergo irreversible reactions with ^1^O_2_, leading to the creation of stable endoperoxides. For instance, ADPA reacts with ^1^O_2_ to produce endoperoxides, with a peak photobleaching absorption occurring at around 378 nm [68]. Absorption probes enable real-time visualization and monitoring of ^1^O_2_, providing a useful tool for exploring the generation, distribution, and dynamics of ^1^O_2_ in biological environments.

#### 4.2.4. Chemiluminescence

Fluorescence photometry and spectrophotometry are based on the reaction of probe molecules with ^1^O_2_, which results in a significant increase in fluorescence or a decrease in absorbance. On the other hand, chemiluminescence does not require an excitation light source and can directly eliminate the interference of background light, making it a suitable method for detecting ^1^O_2_. When the chemical probe reacts with ^1^O_2_, it produces highly energetic compounds, which are usually unstable and release energy during photolysis. The presence or absence of ^1^O_2_ is determined by detecting light at a specific wavelength. Some common chemical probes include 2-methyl-6-phenyl-3,7-dihydroimidazo[1,2-α]-pyrazine-3-one (CLA) [69] and its derivatives, such as 4,4′(5′)-bis[2-(9-anthryloxy)ethyl-thio]tetra-fulvalene (TTF). In the H_2_O_2_/ClO^−^ system, the sensitivity of the detection of ^1^O_2_ is as low as 76 nM [70].

## 5. Available Singlet Oxygen Data of Regulatory-Approved PS

Quantum yield is an important parameter that measures the ability of a PS to produce ^1^O_2_. It is crucial to accurately determine the ^1^O_2_ quantum yield of a particular PS in various media. When an excited PS is quenched by ^3^O_2_, it produces ^1^O_2_. The efficiency of this process is referred to as the ^1^O_2_ quantum yield (Φ_∆_), which is the ratio between the number of ^1^O_2_ produced and the number of photons absorbed by the PS.
(2)$$\Phi_{\Delta }=\frac{\mathrm{number}\;\mathrm{of}\;\sin\mathrm{glet}\;\mathrm{oxygen}\;\mathrm{generated}}{\mathrm{number}\;\mathrm{of}\;\mathrm{photons}\;\mathrm{absorbed}}$$

The Φ_∆_ of a PS is an inherent characteristic that varies depending on the molecular structure and photophysical properties of the PS. Accurately quantifying the Φ_∆_ of a Type II PS is important for evaluating PDT effectiveness [71].

Oxygen depletion is an indirect technique to accurately measure the Φ_∆_ by dissolving the PS in a solvent and passing oxygen through the solution utilizing a gas pump to ensure a steady supply of dissolved oxygen during the reaction. Oxygen consumption is accurately measured under steady-state irradiation conditions using gas microtubes; it is a process that requires sufficient substrate to quantitatively capture and intercept all ^1^O_2_ produced during the photoreaction. The Φ_∆_ can be determined from the ratio between the number of moles of oxygen consumed and the number of einsteins absorbed by the PS [72]. Lysozyme inactivation can be used as an indirect method for measuring the Φ_∆_. When ^1^O_2_ reacts with lysozyme, it leads to the inactivation of lysozyme, and this inactivation is proportional to the concentration of ^1^O_2_. The production of ^1^O_2_ can be determined by measuring the activity of the remaining lysozyme. The Φ_∆_ can be calculated by comparing the difference between lysozyme and lysozyme-free activities [73]. In addition, the excited-state PS is a common precursor of $O_{2}^{•-}$ and ^1^O_2_. The $O_{2}^{•-}$ reacts with cytochrome C, leading to a cytochrome C reduction. This process can be measured by checking the production of $O_{2}^{•-}$ and the number of photons absorbed by the PS, which helps calculate the quantum yield of $O_{2}^{•-}$. The production of ^1^O_2_ also depends on the number of photons absorbed by the PS. Thus, the Φ_∆_ can be indirectly determined by comparing the relative yields of $O_{2}^{•-}$ and ^1^O_2_ [74].

The time-resolved infrared luminescence (TRIL) method can also be used to directly detect the Φ_∆_ generated by a PS. First, the absorption spectrum of the PS is recorded using a spectrophotometer to determine the absorption coefficient. Then, the luminescence intensity of ^1^O_2_ is directly detected using a PMT. By measuring the number of photons absorbed by the PS and the luminescence intensity of ^1^O_2_ the Φ_∆_ can be calculate [75]. These methods are often calibrated using the quantum yield of a reference PS. The most commonly used calibration PSs are Rose Bengal (RB: Φ_∆_ = 0.75) and Methylene Blue (MB: Φ_∆_ = 0.52) [75]; Pheophrbide-a is also used as a calibration reference by some researchers [17]. Table 3 provides a summary of the methods used for the determination of quantum yield and specific values for several regulatory-approved PSs for clinical applications. It should be noted that many direct and indirect measurements are often carried out in deuterated solvents, which can improve ^1^O_2_ detection but might increase experiment costs and complications depending upon the required solvent [75,76].

Based on Table 3, it is evident that various PSs exhibit unique optical properties. The maximum absorption wavelengths (*λ_max_*) and molar extinction coefficients (*Ɛ_max_*) of these PSs can impact their light absorption efficiency and tissue penetration depth, thereby influencing the effectiveness of PDT. The development and design of new PSs of different chemical, biological, and optical properties continue to be the focus of PDT research.

The extinction coefficient (*Ɛ*) quantifies how efficiently a PS absorbs light at a specific wavelength (*λ*). The subscript “max” denotes the peak extinction coefficient (*Ɛ_max_*) and peak absorption wavelength (*λ_max_*), respectively. The penetration depth of light in a target tissue depends on the scattering and absorption properties of the tissue at the specific wavelength of the light [81]. The actual “phototherapeutic window” of PDT ranges between 600 nm and 800 nm [82]. The *λ_max_* and *Ɛ_max_* of porphyrin PSs are relatively low compared with those of other types of PSs. For example, Photofrin^®^, the drug form of Porfimer Sodium, has an *Ɛ_max_* of 3000 M^−1^cm^−1^ at 630 nm, while ALPcS4 has an *Ɛ_max_* of 200,000 M^−1^cm^−1^ at 676 nm [73]. The low *Ɛ_max_* value of the PS at *λ_max_* indicates that the PS absorbs light weakly at the *λ_max_*. Generally speaking, the larger the *Ɛ_max_* value, the higher potential PDT response. However, in contrast, the lower *Ɛ_max_* value of a certain PS does not necessarily correlate to a reduced Φ_∆_. For instance, Photofrin^®^ and ALScP4 exhibit Φ_∆_ of 0.89 and 0.38 in phosphate buffer saline (PBS) (pH 7.4) in the presence of 1 vol% Triton X-100 (PBS/TX100), demonstrating a complex interplay between PS’s absorption optical properties, energy transfer, and efficacy in generating ^1^O_2_ [73]. Numerous studies demonstrate that non-halogenated and halogenated BODIPYs (4,4-difluoro-4-bora-3a,4a-diaza-s-indacene) have very similar *λ_max_* values, but very different Φ_∆_ values, suggesting no absolute correlation between Φ_∆_ and *λ_max_* [83].

The Φ_∆_ of the PS is influenced not only by their inherent structure but also by the measurement technique, ambient environmental conditions, and excitation wavelength. The Φ_∆_ for the same PS, measured under consistent conditions, may differ by measurement methods (e.g., the Φ_∆_ values for PpIX in PBS/TX100 solution are 0.8 and 0.56 when measured using TRIL and Lysozyme inactivation methods, respectively) [73,75]. The Φ_∆_ for a PS measured by the same method can also vary under different environmental conditions. For example, using the oxygen depletion method, the Φ_∆_ values for HpD in methanol and methanol-D solutions are 0.64 and 0.76, respectively [72,77]. Using the TRIL method, the Φ_∆_ values for Talaporfin sodium in Dulbecco’s PBS (D-PBS) and D-PBS/TX100 solutions are 0.53 and 0.59, respectively [73,75]. Even when the same method is used to measure the Φ_∆_ for the same PS under the same solution conditions, there may be some differences. This could be related to the polarity of the solvent, ionic strength, or the interactions between the PS and solvent molecules.

In clinical applications, the *λ_max_*, *Ɛ_max_*, and Φ_∆_ of the PS can be used as the main criteria for selecting a particular PS. Factors such as the metabolism time of the PS in the body, the duration of the skin photosensitivity response, and the selectivity to the diseased tissue also affect the PDT effect. Porphyrin-type PSs are the most extensively studied, but they have longer skin photosensitivity durations and a lower selectivity for diseased tissues. For instance, Porfimer Sodium, which is purified from HpD, requires a higher drug/light dose during treatment and necessitates 4 to 6 weeks of light avoidance after treatment [84]. Chlorin PSs are widely used clinically due to their strong photosensitivity, high target specificity, and minimal side effects. For example, NPe6 hardly participates in metabolic processes in the body, offering better safety [85]. Phthalocyanine PSs have strong absorption in the “phototherapeutic window”, but they exhibit very low or no absorption in the spectrum of sunlight (400~600 nm), thereby reducing the degree of skin photosensitivity [86]. Nevertheless, the selection and use of PSs also require a comprehensive consideration of these properties, pharmacological and therapeutic strategies, and their impact on therapeutic efficacy and patient safety to achieve optimal PDT outcomes [87,88].

## 6. Conclusive Remarks

Singlet oxygen is molecular oxygen in the excited state and widely exists in nature. The fundamental properties of ^1^O_2_ have been well documented. It is well-known that ^1^O_2_ can react with simple organic molecules, complex biological molecules, and cellular components with high-rate constants. ^1^O_2_ could be generated via various processes. However, triplet sensitization is a very effective way to generate the lowest excited singlet state of oxygen. The term “^1^O_2_” usually refers to the lowest energy excited species (^1^Δ_g_) of molecular oxygen. Triplet states of PS molecules under illumination can undergo energy transfer with oxygen molecules in the ^3^$\sum_{g}^{+}$ state, facilitating the ISC process from the triplet to singlet states of O_2_, producing ^1^O_2_. By understanding and utilizing ^1^O_2_ reactivity, it has been possible to develop unique therapeutic tools that can be precisely employed for destructive purposes via ^1^O_2_-mediated oxidative damages, for example, the medical applications of the photoinduced production of ^1^O_2_ in PDT.

Ever since Photofrin^®^ was officially approved as the first PDT PS drug for the treatment of superficial and solid cancers, the development of new non-toxic PS dyes has been focused on improving PDT efficacy, shortening cutaneous photosensitivity, and expanding indications worldwide. Several PSs have received regulatory approval over the past three decades globally. Currently, PDT mediated by these traditional PSs has been used topically and systemically for the treatment of various cancer, pre-cancer, vascular, and infectious diseases. It should be recognized that each PS has its advantages and disadvantages. Nonetheless, the Type II mechanism and the production of ^1^O_2_ are regarded as the fundamental process in current PDT protocols. PSs should ideally possess a high triplet quantum yield leading to the substantial production of ^1^O_2_ upon irradiation. The research on PS should also comprehensively consider their photophysical properties, biocompatibility, targeting, and metabolic pathways to ensure the safety and efficacy of clinical PDT application. It should also be noted that PDT is less effective in hypoxia conditions, since ^1^O_2_ generation in the PDT process requires oxygen and the process also consumes oxygen; therefore, PDT effectiveness can be directly affected by the tissue oxygen partial pressure and oxygenation status. Since the severity of PDT-induced photocytotoxicity is governed by the temporal and spatial distribution of ^1^O_2_, the overproduction of ^1^O_2_ in normal tissue during PDT treatment could cause unwanted collateral damage and side effects.

However, due to the technique difficulty in the direct detection of ^1^O_2_ and the implementation of ^1^O_2_ dosimetry, quantifying the in vivo production of ^1^O_2_ is often an overlooked factor in PS evaluation and PDT dosimetry. In clinical practice, the *λ_max_*, *Ɛ_max_*, and Φ_∆_ of PS can be used as the main criteria for selecting a particular PS. The Φ_∆_ is often used for the evaluation of the ability of a PS to produce ^1^O_2_. It should be noted that, as described above, the reported Φ_∆_ values of regulatory-approved PS are often determined in organic solvents. Even measured under the same condition, the Φ_∆_ for the same PS may differ by measurement methods. Although the direct detection of ^1^O_2_ and Φ_∆_ is more technically challenging within biological environments and particularly in clinical setup, the quantitative techniques for ^1^O_2_ measurement during photosensitization are of immense importance of values for both preclinical and clinical evaluation of potential PSs for future clinical use, for example, the direct singlet oxygen luminescence dosimetry (SOLD). In the scenarios where direct ^1^O_2_ luminescence detection is difficult, the macroscopic singlet oxygen explicit dosimetry (SOED), which involves the measurement of the key components in the PDT photosensitizer concentration, ground-state oxygen concentration ([^3^O_2_]), and light fluence to calculate the amount of reacted ^1^O_2_, might offer a viable alternative for quantifying ^1^O_2_ in PDT.

## Figures and Tables

**Figure 1 pharmaceuticals-17-01274-f001:**
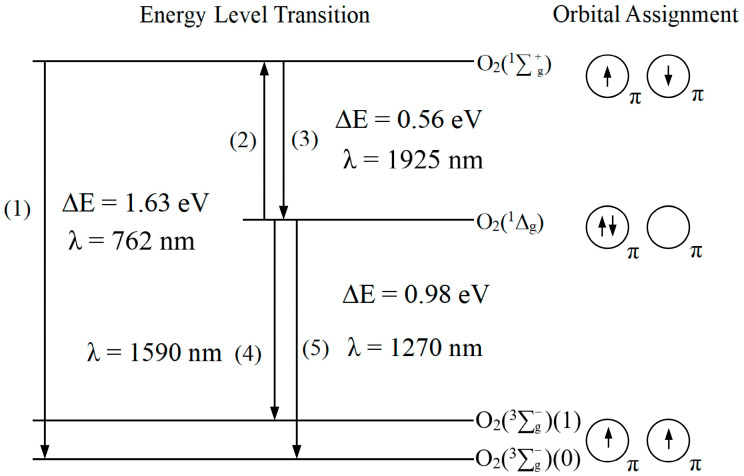
The lowest electronic state energy level transitions and molecular orbital schematics of the oxygen molecule. The configuration of the molecular orbitals of the ^1^Δ_g_ can be described as follows: O_2_KK(2σ_g_)^2^(2σ_u_)^2^(3σ_g_)^2^(1π_u_)^4^(${1\pi }_{g}^{+}$)(${1\pi }_{g}^{+}$).

**Figure 2 pharmaceuticals-17-01274-f002:**
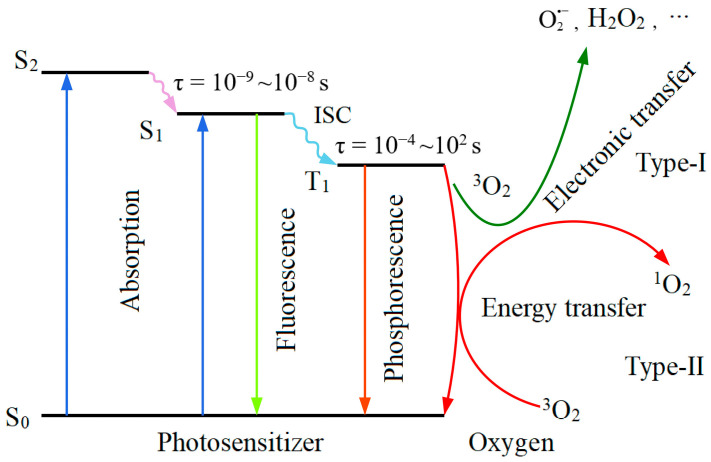
Simplified Jablonski energy level diagram of Type I and Type II photochemical reaction.

**Table 2 pharmaceuticals-17-01274-t002:** Quenching rate constant and quenching mode of commonly used ^1^O_2_ quenchers.

Photosensitizer	Quencher	Solvent	*k_q_ *(M^−1^s^−1^)	Type of Quenching	Reference
Chlorophyll	*β*-Carotene (C_40_H_56_)	Benzene	1.3 × 10^10^	Physical	[25]
		Methanol	9.3 × 10^9^	Physical	
Rose Bengal and Eosin Y	Sodium azide (NaN_3_)	Water	6 × 10^8^	Physical	[26]
2-Acetonaphthone	Vitamin E (α-Tocopherol)	Methanol	3 × 10^8^	Physical	[27]
		Toluene	2.2 × 10^8^	Physical	
Rose Bengal and Porphyrin	Histidine (C_6_H_9_N_3_O_2_)	Water	4.6 × 10^7^	Physical and Chemical	[28]
Rose Bengal and Porphyrin	Tryptophan	Water	3.2 × 10^7^	Physical and Chemical	[28]
Methylene Blue	Tyrosine	water	7 × 10^6^	Chemical	[29]

**Table 3 pharmaceuticals-17-01274-t003:** Characterization of photosensitizer approved for PDT.

Photosensitizer	*λ_max_* (nm)/ *Ɛ_max_* (M^−1^cm^−1^)	Solvent/(Standard)	Φ_∆_	Method	Reference
Porfimer Sodium	632/3000	PBS/TX100 (RB:0.75)	0.89	Lysozyme inactivation	[73]
Hematoporphyrin derivative (HpD)	630	Methanol	0.64	Oxygen depletion	[77]
		Methanol-D	0.76	Oxygen depletion	[72]
Protoporphyrin IX (PpIX)	630/3480	D-PBS/TX100 (RB:0.75)	0.78	TRIL	[75]
		PBS/TX100 (RB:0.75)	0.8	TRIL	[75]
		PBS/TX100 (MB:0.52)	0.56	Lysozyme inactivation	[73]
Meta-tetra(hydroxyl pheny) chlorin (m-THPC)	652/35,000	Ethanol (Pheophorbide-a:0.52)	0.42	TRIL	[78]
N-aspartyl chlorin e6 (NPe6)	664/40,000	D_2_O (RB:0.75)	0.66	Oxygen depletion	[79]
		PBS (MB:0.52)	0.64	Lysozyme inactivation	[80]
Benzoporphyrin derivative monoacid ring A (BPD-MA)	689/34,000	Ethanol	0.81	Cytochrome C reduction	[74]
		Methanol	0.78	Cytochrome C reduction	[74]
		Benzene	0.77	TRIL	[80]
Talaporfin sodium (mono-L-aspartyl chlorine e6)	654/40,000	D-PBS (RB:0.75)	0.53	TRIL	[75]
		D-PBS/TX100 (RB:0.75)	0.59	TRIL	[73]
Aluminum phthalocyanine tetrasulfonate (ALPcS4)	676/200,000	PBS/TX100	0.38	Lysozyme inactivation	[73]

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
