# Peer review of "Singlet Oxygen in Photodynamic Therapy"

_pharmaceuticals, 2024, doi:10.3390/ph17101274_

Round 1

Reviewer 1 Report

Comments and Suggestions for Authors

Referee report for the review article
"Singlet Oxygen in Photodynamic Therapy"
by Shengdong Cui et al.

The article is well written and readable.
There are a few smaller issues, which the authors should improve before publication.
The authors should elaborate more about the theratpeutic mechanism of
singlet-oxygen. The should give a clear statement, why is it good to produce
singlet-oxygen in PDT.
The authors should also reflect on possible disadvantages and side effects.

Page 1, line 45: the expression "based on available literatures" is a bit unusual (literature should be used in the singular).

Page 2, lines 64-66: the authors should explain the differences between reactions
(4) and (5).

Page 2, Figure 1: the authors could improve the discription of the Delta_g state:
for example as a linear combination of pi_x^2 and pi_y^2.

After these minor modifications the paper can be published.

Author Response

We truly appreciate the reviewers’ thoroughness. We made revision based on the reviewers’ and editor’s suggestions and comments.

Comments 1: [The authors should elaborate more about the theratpeutic mechanism of singlet-oxygen. The should give a clear statement, why is it good to produce singlet-oxygen in PDT. The authors should also reflect on possible disadvantages and side effects.]

Response:

Thanks for the suggestion. Following explanation is added to the Introduction to replace “This process can cause oxidative damage to cells to achieve therapeutic outcomes.”:

“The mechanism of singlet oxygen mediated PDT highly depends on the subcellular localization of PS and the interaction between singlet oxygen and organelles such mitochondria, lysosomes, endoplasmic reticulum, Golgi apparatus and plasma membranes. When exceed cell’s antioxidation capacity the acute oxidative stress to these organelles can induce changes in calcium and lipid metabolism, production of cytokines and stress proteins, and activation of protein kinases and transcription factors. These cellular responses can ultimately lead to apoptosis and necrosis of targeted eukaryotic cells and prokaryotic cells. Subsequently, the therapeutic mechanism of singlet oxygen generated cytotoxicity include direct killing of target cell, damaging to blood vessel and inducing immune effect.” 

To answer “why is it good to produce singlet-oxygen in PDT”, following statement is added:

“The triplet sensitization in PDT process is a very effective way to generate 1O2, which makes 1O2 mediated PDT a powerful treatment modality with unique mechanisms and broad clinical applications”

To address possible disadvantages and side effects, following statement is added to the Conclusive remarks:

“It should also be noted that PDT is less effective in hypoxia condition since 1O2 generation in PDT process requires oxygen and the process also consumes oxygen, therefore, PDT effectiveness can be directly affected by the tissue oxygen partial pressure and oxygenation status. Since the severity of PDT induced photocytotoxicity is governed by the temporal and spatial distribution of 1O2, the overproduction of 1O2 in normal tissue during PDT treatment could cause unwanted collateral damages and side effects.”

Comments 2: [Page 1, line 45: the expression "based on available literatures" is a bit unusual (literature should be used in the singular).]

Response:

Agree. Changed to “based on recent literature”.

Comments 3: [Page 2, lines 64-66: the authors should explain the differences between reactions (4) and (5).]

Response:

Thanks for the comment. The difference between the reaction or transition (4) and (5) is the wavelength of emit light. The sentence and Figure 1 are rearranged to indicate the specific wavelength.  

Comments 4: [Page 2, Figure 1: the authors could improve the discription of the Deltag state: for example as a linear combination of pi_x^2 and pi_y^2.]

Response:

Agree. We added the description O2(1Δg) in Figure 1 and modified to read “The lowest electronic state energy level transitions and molecular orbital schematics of oxygen molecule. The configuration of the molecular orbitals of the 1Δg, is as follows: O2KK(2σg)2(2σu)2(3σg)2(1πu)4(1π+ g) (1π+ g)”.

Reviewer 2 Report

Comments and Suggestions for Authors

Comments to the Author
The authors of the manuscript pharmaceuticals-3157558-peer-review-v1, presented an interesting review article about the properties, production of singlet Oxygen 1O2 in Photodynamic Therapy, and the possible detection methods. The topic is interesting and suited to the Journal topic. Yet, the manuscripts need improvements in the graphical illustrations that make the manuscript easy to follow, particularly in sections 2.3. Oxidizing Activity of Singlet Oxygen, 4. Singlet Oxygen Detection and 5. Available Singlet Oxygen Data of Regulatory Approved PS. In addition, some minor revision/typos should be revised also

-            In subsection “2.1. Electronic Structure and Leaps in Singlet Oxygen”.. please relace the word “reaction” with transition…for example reaction (5) revised to transition (5).

-            I recommend the author to include Type photoreaction transition in Fig.2. and refer to the difference between the two process (type I and Type II)

-            The section “3. Singlet Oxygen Production in PDT Process” is very short and authors did not give review to the various materials that were used to generate the singlet oxygen in the PDT process.

Author Response

We truly appreciate the reviewers’ thoroughness. We made revision based on the reviewers’ and editor’s suggestions and comments.

The authors of the manuscript pharmaceuticals-3157558-peer-review-v1, presented an interesting review article about the properties, production of singlet Oxygen 1O2 in Photodynamic Therapy, and the possible detection methods. The topic is interesting and suited to the Journal topic. Yet, the manuscripts need improvements in the graphical illustrations that make the manuscript easy to follow, particularly in sections 2.3. Oxidizing Activity of Singlet Oxygen, 4. Singlet Oxygen Detection and 5. Available Singlet Oxygen Data of Regulatory Approved PS. In addition, some minor revision/typos should be revised also

Comments 1: [In subsection “2.1. Electronic Structure and Leaps in Singlet Oxygen”. please replace the word “reaction” with transition…for example reaction (5) revised to transition (5).]

Response:

Agree. Changed “reaction” to “transition”.

Comments 2: [I recommend the author to include Type photoreaction transition in Fig.2. and refer to the difference between the two process (type I and Type II)]

Response:

Thanks for the suggestion. Type I reaction is added to Figure 2.

Comments 3: [The section “3. Singlet Oxygen Production in PDT Process” is very short and authors did not give review to the various materials that were used to generate the singlet oxygen in the PDT process.]

Response:

Thanks for the suggestion. Following paragraphs are added:

“PDT is a two-step procedure: the patient is first topically or systemically given a PS (“photosensitizing drug”) that preferentially accumulates in the target tissue. Light of a specific wavelength generated by laser light source, LED light source or daylight is then applied to the treatment site in order to active the PS. The process of PDT to produce 1O2 involves the interaction of PS molecules with light irradiation and molecular oxygen. Therefore, PS, light and oxygen are considered as the three crucial elements of PDT. The process of PDT to produce 1O2 includes photon absorption, emission, and energy transfer as PS undergoes different electronic states to eventually generate an excited triplet state PS and convert the absorbed light energy to highly cytotoxic 1O2. PSs are usually conjugated unsaturated organic molecules that possess high light absorption coefficients, high intersystem crossing efficiencies and high quantum yields for the production of 1O2 or other ROS.”

Reviewer 3 Report

Comments and Suggestions for Authors

Huang and co-worker present in their submission to Pharmaceuticals "Singlet Oxygen in Photodynamic Therapy". This is a generally well written review with many interesting aspects of the physical chemical of singlet oxygen. The following issues mentioned below must be addressed.

Major:

"Contrary to what might be believed, the shorter maximum absorption wavelengths and lower Ɛmax of a certain PS do not necessarily correlate with a reduced ΦΔ value. For instance, Photofrin® and ALScP4 exhibit ΦΔ of 0.89 and 0.38 in phosphate buffer in the presence of 1 vol% Triton X-100 (PB/TX100), demonstrating a complex interplay between a PS’s absorption optical properties and its efficacy in generating singlet oxygen [70]." The referee would argue that there is absolutely no correlation between absorption maximum and singlet oxygen quantum yield. Consider for this iodinated and non-iodinated BODIPYs (they have a very similar absorption, but very different singlet oxygen quantum yields).

It should be mentioned that the absence of oxygen in hypoxic tumor tissue and thereby also the reduced singlet oxygen concentration will reduce the efficiency of PS in PDT.

Minor:

"Aminolevulinic acid (ALA) and its ester derivatives, a prodrug that is metabolized in cells and enzymatically converted to protoporphyrin IX (PpIX), a potent endogenous porphyrin type PS [4]." The main sentence lacks a verb.

"(penTafluorophenyl) porphine (TPPF)" should be "(pentafluorophenyl) porphine (TPPF)"

"β-Caritene" shouldn't this be "β-Carotene"?

"Reacts readily with unsaturated organic compounds by electrophilic addition and electron extraction [11]." This sentence lacks a noun.

"For instance, the half-life of 1O2 is approximately 10-6 s and that of •OH is 10-9 s [41]." These values apply for which type of solution (pure water, PBS etc.)?

Chapter 4.1: It should be mentioned that for improved singlet oxygen detection, these measurements are frequently done in deuterated solvents, which can become costly depending upon the required solvent (see e.g. 10.1016/j.jphotobiol.2007.02.006, 10.1007/978-3-031-02391-0_4, 10.1021/acs.jmedchem.1c00052 and 10.1021/acs.chemrev.4c00105).

"as low as 76 nM/L" should be "as low as 76 nM"

"PB" is a rather uncommon abbreviation, its meaning should be explained.

References for the singlet oxygen quantum yield standards (RB, MB, Pheophorbide-a) should be given.

The numbering of the page numbers in the references is inconsistent: compare "459-64" with "343-344".

Ref. 50: First and family name have been exchanged for one author. The page number is missing.

Ref. 60 is a book chapter from the publisher IntechOpen, not from a journal called Biochemistry.

Ref. 65 has a journal name, which is not abbreviated in contrast to the rest of the manuscript.

Ref. 69 has a wrongly abbreviated journal name.

Comments on the Quality of English Language

See my report

Author Response

"Contrary to what might be believed, the shorter maximum absorption wavelengths and lower Ɛmax of a certain PS do not necessarily correlate with a reduced ΦΔ value. For instance, Photofrin® and ALScP4 exhibit ΦΔ of 0.89 and 0.38 in phosphate buffer in the presence of 1 vol% Triton X-100 (PB/TX100), demonstrating a complex interplay between a PS’s absorption optical properties and its efficacy in generating singlet oxygen [70]." The referee would argue that there is absolutely no correlation between absorption maximum and singlet oxygen quantum yield. Consider for this iodinated and non-iodinated BODIPYs (they have a very similar absorption, but very different singlet oxygen quantum yields).

Response:

Thanks for the comments. We rewrite the paragraph and also add the reviewer’s point:

“The low Ɛmax of PS at λmax indicates that the PS absorbs light weakly at the λmax. Generally speaking, the larger Ɛmax the higher potential PDT response. However, in contrary, the lower Ɛmax of a certain PS do not necessarily correlate to a reduced Φ. For instance, Photofrin® and ALScP4 exhibit Φ of 0.89 and 0.38 in phosphate buffer saline (PBS) (pH 7.4) in the presence of 1 vol% Triton X-100 (PBS/TX100), demonstrating a complex interplay between PS’s absorption optical properties, energy transfer and efficacy in generating 1O2 [70]. Numerous studies demonstrate that halogenated and regioisomeric biphenyl BODIPYs (4,4-difluoro-4-bora-3a,4a-diaza-s-indacene) have very similar λmax but very different Φ, suggesting no absolute correlation between Φ and λmax“.

Comments 1: [It should be mentioned that the absence of oxygen in hypoxic tumor tissue and thereby also the reduced singlet oxygen concentration will reduce the efficiency of PS in PDT.]

Response:

Thanks for the suggestion. This limitation has been added to the Conclusive remarks.

Comments 2: ["Aminolevulinic acid (ALA) and its ester derivatives, a prodrug that is metabolized in cells and enzymatically converted to protoporphyrin IX (PpIX), a potent endogenous porphyrin type PS [4]." The main sentence lacks a verb.]

Response:

Agree. Correction is made by adding “are”.

Comments 3: ["(penTafluorophenyl) porphine (TPPF)" should be "(pentafluorophenyl) porphine (TPPF)"]

Response:

Agree. Correction is made by changing “T” to “t”.

Comments 4: [ "β-Caritene" shouldn't this be "β-Carotene"?]

Response:

Agree. Corrections are made.

Comments 5: [ "Reacts readily with unsaturated organic compounds by electrophilic addition and electron extraction [11]." This sentence lacks a noun.]

Response:

Agree. Correction is made by adding “singlet oxygen”.

Comments 6: [ "For instance, the half-life of 1O2 is approximately 10-6 s and that of •OH is 10-9 s [41]." These values apply for which type of solution (pure water, PBS etc.)?]

Response:

The original article and cited references did not specify what type of solvent was used. We changed the statement to”

“This can also be reflected by the difference between the wide ranges of half-life of different ROS. For instance, the half-life of 1O2 ranges between 10-9 – 10-6 s and that of OH between 10-9 – 10-7 s ”.

Comments 7: [ Chapter 4.1: It should be mentioned that for improved singlet oxygen detection, these measurements are frequently done in deuterated solvents, which can become costly depending upon the required solvent (see e.g. 10.1016/j.jphotobiol.2007.02.006, 10.1007/978-3-031-02391-0_4, 10.1021/acs.jmedchem.1c00052, and 10.1021/acs.chemrev.4c00105).]

Response:

Thanks for the suggestion. Following comment and references are added to the Section 5 and after the Table 3 since D2O is used in some references:

“It should be noted that many direct and indirect measurements are often carried out in deuterated solvents, which can improve 1O2 detection but increase experiment cost and complication depending upon the required solvent”.

Comments 8: [ "as low as 76 nM/L" should be "as low as 76 nM"]

Response:

Agree. We have changed “76 nM/L” to “76 nM”

Comments 9: ["PB" is a rather uncommon abbreviation, its meaning should be explained.]

Response:

Agree. Proper abbreviation should be “PBS”. “PB” is replaced by “PBS”.

Comments 10: [References for the singlet oxygen quantum yield standards (RB, MB, Pheophorbide-a) should be given.]

Response:

Agree. References are added.

Comments 11: [The numbering of the page numbers in the references is inconsistent: compare "459-64" with "343-344".]

Response:

Agree. We have revised the page number format for the references.

Comments 12: [Ref. 50: First and family name have been exchanged for one author. The page number is missing.]

Response:

Agree. Corrections are made.

Comments 13: [Ref. 60 is a book chapter from the publisher IntechOpen, not from a journal called Biochemistry.]

Response:

Agree. We have modified reference 60 to read “Photophysical Detection of Singlet Oxygen. Reactive Oxygen Species 2022, 1-20.”.

Comments 14: [Ref. 65 has a journal name, which is not abbreviated in contrast to the rest of the manuscript.]

Response:

Agree. Corrections are made.

Comments 15: [Ref. 69 has a wrongly abbreviated journal name.]

Response:

Agree. Corrections are made.

Round 2

Reviewer 1 Report

Comments and Suggestions for Authors

The authors have solved all issues.

The manuscript can be published as it is.

Author Response

Thank you for recognizing the article.

Reviewer 2 Report

Comments and Suggestions for Authors

The authors did a real effort to improve the manuscript.  They have answered my inquiries and I accept them.  Therefore, I recommend the last revised version for publication in Pharmaceuticals 

Author Response

Thank you for recognizing the article.

Reviewer 3 Report

Comments and Suggestions for Authors

Huang and co-worker have improved their submission to Pharmaceuticals "Singlet Oxygen in Photodynamic Therapy". The following minor issues mentioned below must be addressed and then publication can occur.

"Numerous studies demonstrate that halogenated and regioisomeric biphenyl BODIPYs (4,4-difluoro-4-bora-3a,4a-diaza-s-indacene) have very similar λmax but very different ΦΔ, suggesting no absolute correlation between ΦΔ and λmax [83]." Should be "Numerous studies demonstrate that non-halogenated and halogenated BODIPYs (4,4-difluoro-4-bora-3a,4a-diaza-s-indacene) have very similar λmax but very different ΦΔ, suggesting no absolute correlation between ΦΔ and λmax [83]."

“Aminolevulinic acid (ALA) and its ester derivatives are a prodrug that is metabolized in cells and enzymatically converted to protoporphyrin IX (PpIX), a potent endogenous porphyrin type PS [5].” Should be “Aminolevulinic acid (ALA) and its ester derivatives are prodrugs that are metabolized in cells and enzymatically converted to protoporphyrin IX (PpIX), a potent endogenous porphyrin type PS [5].”

Ref. 63: The publisher should be mentioned here.

Comments on the Quality of English Language

See review.

Author Response

Comment 1

“Numerous studies demonstrate that halogenated and regioisomeric biphenyl BODIPYs (4,4-difluoro-4-bora-3a,4a-diaza-s-indacene) have very similar λmax but very different ΦΔ, suggesting no absolute correlation between ΦΔ and λmax [83]."] Should be “Numerous studies demonstrate that non-halogenated and halogenated BODIPYs (4,4-difluoro-4-bora-3a,4a-diaza-s-indacene) have very similar λmax but very different ΦΔ, suggesting no absolute correlation between ΦΔ and λmax [83].”

Response:

Agree. Changed to “Numerous studies demonstrate that non-halogenated and halogenated BODIPYs (4,4-difluoro-4-bora-3a,4a-diaza-s-indacene) have very similar λmax but very different ΦΔ, suggesting no absolute correlation between ΦΔ and λmax [83].”

Comment 2

“Aminolevulinic acid (ALA) and its ester derivatives are a prodrug that is metabolized in cells and enzymatically converted to protoporphyrin IX (PpIX), a potent endogenous porphyrin type PS [5].” Should be “Aminolevulinic acid (ALA) and its ester derivatives are prodrugs that are metabolized in cells and enzymatically converted to protoporphyrin IX (PpIX), a potent endogenous porphyrin type PS [5].”

Response:

Agree. Changed to “Aminolevulinic acid (ALA) and its ester derivatives are prodrugs that are metabolized in cells and enzymatically converted to protoporphyrin IX (PpIX), a potent endogenous porphyrin type PS [5]”

Comments 3:

Ref. 63: The publisher should be mentioned here.

Response:

Thanks for the reminder. Full citation is added following the journal’s Reference Formatting Guide:

Maity, A. Photophysical Detection of Singlet Oxygen In Reactive Oxygen Species; Ahmad, R. and Surguchov, A. Ed.; IntechOpen: London, UK, 2022; Chapter 2, pp. 1-20. [Internet]